# Metronomic Chemotherapy Based on Topotecan or Topotecan and Cyclophosphamide Combination (CyTo) in Advanced, Pretreated Ovarian Cancer

**DOI:** 10.3390/cancers15041067

**Published:** 2023-02-07

**Authors:** Piotr J. Wysocki, Mateusz Łobacz, Paweł Potocki, Łukasz Kwinta, Anna Michałowska-Kaczmarczyk, Agnieszka Słowik, Kamil Konopka, Anna Buda-Nowak

**Affiliations:** 1Department of Oncology, Jagiellonian University—Medical College Hospital, 31-501 Cracow, Poland; 2Department of Oncology, Jagiellonian University—Medical College, 31-008 Cracow, Poland

**Keywords:** ovarian cancer, metronomic chemotherapy, salvage treatment, topotecan, topotecan + cyclophosphamide, BMI

## Abstract

**Simple Summary:**

In this retrospective analysis of 72 advanced ovarian cancer patients, we have evaluated the safety and activity of orally-administered metronomic chemotherapy (MC) based on single agent topotecan or a combination of topotecan and cyclophosphamide (CyTo regimen). In this difficult population, metronomic chemotherapy demonstrated a clinically meaningful activity and good safety profile. The MC provided a clinical benefit in the majority of treated patients, with fewer than 15% not benefitting from this treatment (biochemical or radiographic progression). The median PFS in the whole population was 3.65 months, but the median PFS in patients with a biochemical response to MC (18.2% of patients) reached 10.7 months. The study also showed that overweight or obese patients had significantly better outcomes on MC than patients with BMI <25 kg/m^2^. This analysis established the CyTo regimen as the preferred MC to be evaluated in a phase II clinical trial currently under construction.

**Abstract:**

Patients with advanced ovarian cancer (OC) have a detrimental prognosis. The options for systemic treatment of advanced OC in later lines of treatment are limited by the availability of active therapies and their applicability to often fragile, exhausted patients with poor performance status. Metronomic chemotherapy (MC) is a concept of a continuous administration of cytotoxic drugs, which is characterized by multidirectional activity (anti-proliferative, anti-angiogenic, and anti-immunosuppressive) and low toxicity. We have performed a retrospective analysis of consecutive, advanced, chemo-refractory OC patients treated with MC based on single-agent topotecan (1 mg p.o. q2d) or on a topotecan (1 mg q2d) and cyclophosphamide (50 mg p.o. qd) combination (CyTo). Metronomic chemotherapy demonstrated promising activity, with 72% and 86% of patients achieving biochemical or objective disease control and 18% and 27% of patients achieving a biochemical or objective response, respectively. The median PFS in the whole population was 3.65 months, but the median PFS in patients with a biochemical response to MC (18.2% of patients) reached 10.7 months. The study also suggested that overweight or obese patients had significantly better outcomes on MC than patients with BMI <25 kg/m^2^. This article is the first report in the literature on metronomic chemotherapy based on a topotecan + cyclophosphamide combination (CyTo). The CyTo regimen demonstrated safety, clinical activity, and potential broad clinical applicability in advanced OC patients and will be evaluated in a forthcoming clinical trial.

## 1. Introduction

According to Globocan’s 2020 projections, by 2040, the number of women around the world diagnosed with ovarian cancer will increase by 42% to 445,721 [1]. The number of women dying from ovarian cancer each year is projected to increase from 2020 by over 50% to 313,617. Despite significant improvements in the diagnosis and treatment of this disease, the prognosis of patients diagnosed with ovarian cancer remains poor. Less than 35% of patients diagnosed with this disease (313,959 new cases worldwide) can be possibly cured, which makes >65% of patients (207,252 deaths) who will ultimately develop an advanced, incurable disease and will require long-term palliative systemic therapy [1]. Ovarian cancer incidence is strongly related to age, with the highest incidence rates being in older women. In the UK in 2016–2018, on average each year, more than a quarter of new cases (28%) were in females aged 75 and over [2]. There is no doubt, that older patients, especially those with coexisting comorbidities, are at higher risk of treatment-related adverse events leading to treatment interruption, disease progression, and death [3,4]. However, even in younger patients, many standard chemotherapy regimens are associated with significant toxicity, which can impair treatment intensity, preclude prolonged, systemic maintenance treatment, and simultaneously significantly deteriorate patients’ quality of life [5]. Moreover, there is still insufficient progress in the quest for novel agents with significant activity in advanced OC patients both at early and late lines of therapy. Treatment of recurrent OC is based on the paradigm of platinum sensitivity where platinum-sensitive patients are re-treated with platinum-based therapy, and platinum-resistant patients receive monotherapy or combinations of non-platinum agents (pegylated liposomal doxorubicin, paclitaxel, carboplatin, gemcitabine, topotecan, cyclophosphamide) [6,7]. However, in many cases of recurrent disease, treatment options are limited due to the persistence of adverse events such as neuropathy, nephrotoxicity, or myelosuppression. Due to such circumstances, later lines of intensive systemic treatment become problematic, and patients are treated with few available, relatively low-toxic but also low-active cytotoxic drugs usually administered as monotherapy [8]. One of the therapeutic options that can be considered in the late lines of therapy is metronomic chemotherapy based on very frequent administration of orally available cytotoxic agents at low doses. We have decided to retrospectively evaluate the safety and efficacy of metronomic chemotherapy (MC) administered to pretreated, advanced OC patients. The analysis confirmed the safety of MC in such patients and provided promising clinical signals of MC’s activity in ovarian cancer.

## 2. Materials and Methods

### 2.1. Patients

We retrospectively collected data on advanced OC patients treated with metronomic chemotherapy at Jagiellonian University-Medical College Hospital in Cracow between 2017 and 2022. Metronomic chemotherapy has been initially offered to pretreated, progressing (radiographic and biochemical progression) patients who were deemed unfit or refused further lines of standard intravenous chemotherapy. Subsequently, the treatment was also offered to pretreated, progressing patients who were asymptomatic or mildly symptomatic and were reluctant to initiate intensive, intravenous chemotherapy immediately. The eligible patients had histologically proven advanced (metastatic or locally recurrent) inoperable ovarian cancer and presented with a performance status of ECOG 0–2. The study has been approved by the local bioethical committee at Jagiellonian University

### 2.2. Treatment

The metronomic chemotherapy offered to OC patients evolved over time, with an initial treatment consisting of metronomic single-drug topotecan administered at a dose of 1 mg every second day (q2d) p.o. Upon asymptomatic, biochemical progression, patients who had experienced earlier at least disease stabilization on topotecan monotherapy received a combined regimen (CyTo) consisting of topotecan 1 mg q2d and cyclophosphamide (50 mg once daily (qd) p.o.). The safety and feasibility of the topotecan-based regimen were previously described [9,10,11]. The decision to combine topotecan with cyclophosphamide was based on our experience with metronomic cyclophosphamide in breast cancer patients [data submitted]. Upon confirmation of the safety of the CyTo regimen, all subsequent OC patients were treated with the doublet metronomic chemotherapy.

Data on the following background characteristics of the patients were collected using the standardized data collection instrument: age; ECOG performance status; clinical symptoms; serum tumor markers, including CA-125; tumor stage (locally advanced or metastatic); sites of distant metastases; and pathological diagnosis including immunohistochemistry. As markers of efficacy, we collected data on the objective response, biochemical response, and progression-free survival (PFS).

### 2.3. Analysis of Treatment Efficacy

Since the metronomic treatment was initially used as a last-resort therapy for heavily pretreated OC patients, an objective evaluation of the tumor by computer tomography was conducted only if clinically indicated. Clinical decisions were mainly based on the patient’s performance status and disease-related symptoms as well as on the patient’s preferences. In most cases, the CA-125 serum marker fluctuations were used as a measure of the anti-tumor activity of the metronomic treatment. However, when the clinical benefit of MC became evident, and two-drug MC was used on a more regular basis, more patients were evaluated using advanced imaging modalities (CT, MRI), and responses were evaluated according to RECIST 1.1 criteria.

The biochemical response was analyzed according to Gynecologic Cancer Intergroup (GCIG) guidelines. Briefly, patients eligible for evaluation of their biochemical responses should have had baseline CA-125 level ≥ 2× ULN (upper limit of the norm-35 U/mL). Complete biochemical response (bCR) was defined as CA-125 ≤ ULN maintained for ≥1 month, and partial response (bPR) as ≥ 50% decrease in CA-125 level compared to baseline. Biochemical progression (bPD) was defined as a CA-125 level ≥ 70 U/mL (if baseline or nadir ≤ULN) or ≥ 2× increase in CA-125 level (if baseline or nadir ≥ ULN). Biochemical stabilization (bSD) was defined as a CA-125 level not fulfilling criteria for bCR, bPR, or bPD, which was maintained for ≥1 month.

Progression-free survival (PFS) was analyzed in evaluable patients receiving MC. However, in order to avoid a bias, the comparison of PFS between both MC regimens excluded patients treated initially with topotecan monotherapy who switched to topotecan + cyclophosphamide combination upon disease progression.

### 2.4. Safety Analysis

Data on treatment-related myelotoxicity was obtained by automatic analysis of laboratory results of blood samples collected every four weeks during treatment with the FulVEC regimen. Data on other, bone marrow-unrelated AEs and on FulVEC dosage modification or interruption was derived directly from patients’ medical history.

### 2.5. Statistical Considerations

Distributions of quantitative variables were summarized with mean, standard deviation, median, and quartiles, whereas distributions of qualitative variables were summarized with the number and percent of occurrence for each of their values.

Logistic regressions were used to analyze the impact of quantitative variables on dichotomous outcomes. Odds ratios (OR) with 95% confidence intervals were shown.

Kaplan–Meier curves were compared with LR (log-rank) test. The significance level for all statistical tests was set to 0.05. R 4.2.1. was used for computations.

## 3. Results

Between September 2017 and June 2022, 72 patients received palliative topotecan-based metronomic chemotherapy. Topotecan monotherapy (T) was administered to 18 patients (25%) and topotecan + cyclophosphamide (CyTo) to 45 patients (62.5%). Nine patients (12.5%) who initially received topotecan were treated with CyTo combination upon disease progression. The median follow-up was 8.57 months. A detailed clinicopathological characteristic of the patients is included in Table 1.

Objective tumor response could be assessed in 22 evaluable patients who underwent regular CT-based imaging. Objective responses were observed in six patients (27.2%) with one complete (CR) and five partial responses (PR). Clinical benefit rate (CBR) involving CR, PR, and disease stabilization was observed in 19 patients (86.3%). Only three evaluable patients (13.6%) experienced progressive disease as the best response [Figure 1a].

Thirty-seven patients were evaluable for assessment of biochemical response. Five patients (11.4%) experienced a complete response and three patients (6.8%) partial response. Stabilization of the CA-125 level was achieved in 24 patients (54.6%). Only five patients (11.4%) experienced biochemical progression as the best response [Figure 1b]. Surprisingly, biochemical responses were observed only in overweight patients (bCR + bPR = 35.7%).

The median progression-free survival in the group of evaluable (n:69) patients was 3.65 months, and the 3-, 6- and 12-month PFS rate was 62.0%, 26.0%, and 10.1%, respectively, [Figure 2a]. The median PFS of patients treated with topotecan monotherapy (T) or topotecan + cyclophosphamide combination (TC) was 3.45 months, but the 3-, 6-, and 12-month PFS rates were 60.0%, 6.7%, and 0.0%, respectively, in patients treated with T, and 57.2%, 26.7%, and 11.3%, respectively, in patients treated with TC [Figure 2b].

The median PFS was not significantly impacted either by histological tumor type (serous vs. others) [Figure 3a] or grade (low vs. high) [Figure 3b]. However, the 12-month PFS rate in patients with low-grade tumors was 12.8%, whereas in patients with high-grade tumors, it was 4.4%. Serous histology compared to other tumor types was associated with a higher 12-month PFS rate of 13.1% vs. 5.2%.

The number of previous lines of systemic treatment and platinum sensitivity represented factors that significantly impacted the PFS. Median PFS in the platinum-sensitive population was 4.14 months with 3-, 6-, and 12-month PFS rates of 70%, 39.1%, and 20%, respectively, while median PFS in the platinum-resistant patients was 3.32 months with 3-, 6-, and 12-month PFS rates of 55.8%, 15.9%, and 2.7%, respectively, (*p* = 0.01) [Figure 3c]. Median PFS in patients who failed ≤2 lines of chemotherapy was 3.65 months and 3.02 months in patients after >2 lines of treatment (*p* = 0.035) [Figure 3d].

The biochemical response to MC was the most important predictive factor of improved PFS. Patients who experienced bCR or bPR had a median PFS of 10.7 months compared to 4.1 months in patients with bSD or bPD (*p* = 0.002) [Figure 4a].

### 3.1. Outcomes of Overweight Advanced OC Patients

The analysis of the impact of overweight or obesity on the outcomes of MC was conducted in 54 evaluable patients of whom the majority (57.4%) was overweight or obese with BMI >25 kg/m^2^. The median PFS of 4.14 months in overweight patients was significantly longer than the median PFS in patients with BMI < 25 kg/m^2^ (3.12 months—*p* = 0.012). The 3-, 6-, and 12-month survival rate in overweight patients was 60.7%, 37.8%, and 12.4%, whereas in patients with BMI, it was <25–57.1%, 0.0%. and 0.0% [Figure 4b].

### 3.2. Safety

Overall, 64 patients were evaluable for assessment of the safety profile of metronomic chemotherapy. Treatment-related adverse events of any grade were observed in 89.1% of patients, with WHO CTC G3–4 occurring in 32.8% of them [Table 2]. Myelotoxicity represented the most common type of AE, with anemia and neutropenia of any grade and G3–4 being the most frequent among them. Other AEs included increased liver function markers (transaminase and bilirubin) and increased creatinine levels. The CyTo regimen was associated with mild nausea in 14.1% of patients and constipation in 6.3%. Dose reduction at some time points was required in 32 patients (50%) and temporary treatment interruption in 1 patient (1.6%). All treatment-emergent AEs subsided rapidly upon dose reduction. No patients required a permanent cessation of metronomic chemotherapy due to toxicity.

## 4. Discussion

The mainstream of systemic treatment for OC is platinum-based polychemotherapy, which represents the treatment of choice both in radical (neoadjuvant, adjuvant) and palliative settings [12,13,14]. However, truly uncurable, high-risk patients with a measurable disease receiving a standard first-line therapy combining carboplatin, paclitaxel, and bevacizumab have a median PFS merely exceeding one year (14.1–15.9 months) [15,16]. The advent of PARP inhibitors significantly improved outcomes of patients responding to first-line chemotherapy, but approx. 30% of patients will not achieve an objective response qualifying them for this type of treatment [17,18,19]. Nevertheless, all patients will ultimately fail the first line of treatment with or without PARPi, and the activity of further lines of therapy is usually modest and short-lasting. Especially, patients with platinum-resistant disease have detrimental outcomes [7]. With further lines of therapy and repeated incidences of disease relapse, patients’ performance status and quality of life usually significantly deteriorate, thus making the patients simply unfit for the continuation of systemic treatment [8]. Therefore, despite significant progress in the systemic treatment of advanced OC, novel therapeutic approaches are urgently awaited. Such novel approaches may not only include new cytotoxic or molecularly targeted agents but also novel concepts of treatment based on combinations of old drugs, administered in a unique fashion, which exploits their synergistic and multidirectional mechanisms of action.

Metronomic chemotherapy (MCT) represents such a unique approach in that it is a concept of continuous administration of cytotoxic drugs at low doses. Unlike standard chemotherapy regimens in OC, which usually use maximal-tolerated doses (MTD) of chemotherapeutics and require long ≥ 3 weeks recovery periods, the metronomic chemotherapy safety profile allows for continuous, very frequent drug administration [20,21,22]. Unlike MTD-based chemotherapy, which exerts its anti-tumor activity predominantly via interruption of the cell cycle, MC not only inhibits tumor-cell proliferation but also activates other, clinically essential mechanisms, which represent crucial hallmarks of metronomic chemotherapy. These include improvement of anti-tumor immune responses and inhibition of angiogenesis [23,24,25,26]. Owing to its very good safety profile and multidirectional antitumor activity, MC represents a promising but broadly unappreciated option for the systemic treatment of many tumor types, including ovarian cancer [21,25,27,28,29,30]. However, the data on the application of MC in gynecological tumors, and in particular in ovarian cancer, is relatively limited, with only metronomic cyclophosphamide being evaluated in a prospective manner.

In a case report, metronomically administered cyclophosphamide (50 mg qd) to a young patient with chemo-refractory ovarian cancer led to disease stabilization and a biochemical response lasting for >5 years [31]. A retrospective analysis of 68 advanced OC patients treated at a single institution revealed a clinical benefit in 48% of patients with partial response (PR) or disease stabilization (SD) (radiographic/biochemical) in 32% and 16% of patients, respectively [28]. However, most of the patients (52%) did not benefit from the treatment that was characterized by an immediate disease progression. Median PFS and OS in the treated population were 2.6 and 6.0 months, respectively. Another retrospective, multicenter analysis of metronomic cyclophosphamide in 54 pretreated, advanced OC patients revealed a median PFS and OS of 4 and 13 months, respectively. The objective response rate (ORR) in the whole population was 20.4%, with CR in 5.5% and PR in 14.8% of patients, respectively [32]. Clinical benefit (CR, PR, SD) was achieved in 40.7% of patients. Again the majority of patients (59.2%) did not derive any benefit from the treatment. The ORR was numerically higher in platinum-sensitive (23.5%) patients than in platinum-resistant patients (15%). However, the clinical benefit rate was similar between these subgroups—45% and 38% in platinum-sensitive and platinum-resistant patients, respectively. In a prospective clinical study, a combination of metronomic cyclophosphamide with bevacizumab was administered upon progression on bevacizumab in pretreated (≤2 lines) OC patients [33]. The ORR and SD rates were 10% and 65%, with a median PFS of 8.41 months and OS of 22.72 months, respectively. A combination of cyclophosphamide with another antiangiogenic agent (pazopanib) in the treatment of advanced platinum-refractory or resistant OC patients was also prospectively evaluated. Twenty patients received cyclophosphamide 50 mg qd combined with pazopanib 600 mg p.o. qd. The treatment was associated with PR, SD, and PD rates of 45%, 30%, and 25%, respectively. The median PFS and OS were 5.5 months and 9.5 months, respectively [34].

Topotecan is another, orally available, agent with proven activity in advanced OC patients; however, it is routinely administered at the maximum tolerated dose (MTD) but not metronomic doses. Intravenous administration of topotecan was shown to be at least equivalent to paclitaxel in terms of response rate or time-to-progression [35]. The activity of metronomic topotecan in ovarian cancer has been demonstrated mainly in preclinical settings, where this drug displayed strong anti-antiangiogenic potential in a murine orthotopic model of OC [25]. Compared to the administration of topotecan at the maximum tolerated dose (MTD), metronomic administration of this drug resulted in similar anticancer activity with a much better safety profile [9,10,11]. The majority of clinical studies evaluating the role of oral topotecan in the treatment of advanced OC patients utilized this agent at the MTD (2.3 mg/m2/day 1–5 q3w) and demonstrated comparable efficacy of oral and intravenous single-agent topotecan [36,37].

The concept of metronomic chemotherapy combining topotecan and cyclophosphamide (CyTo) has never been previously published. Therefore, this is the first report on the clinical efficacy and safety of this unique regimen. The clinical efficacy of the CyTo regimen compares favorably to previously reported studies on metronomic chemotherapy in OC with relatively higher rates of patients achieving at least disease stabilization (86.3%) and lower rates of patients completely resistant to this treatment (13.6%) [28,32,33]. It must be underscored that the main goal of metronomic treatment is rather a long-lasting control of disease than induction of objective responses, which often requires intensive chemotherapy administered at the MTD. Metronomic chemotherapy is not a treatment for every patient with advanced OC but should be considered an optimal palliative approach in asymptomatic or mildly symptomatic advanced OC patients in the absence of imminent visceral crisis. Low toxicity and convenient dosage allow for long-term, continuous administration of the CyTo regimen with the option for a precise dosage adjustment in the case of treatment-related AE. The lack of patients who permanently ceased the CyTo treatment due to toxicity confirms the applicability of this regimen in everyday practice in advanced OC patients.

One of the crucial hallmarks of metronomic chemotherapy is its safety. The very low number of patients requiring temporary treatment interruption and the lack of patients stopping the treatment due to AE represent a confirmation thereof. A dose-finding study on the topotecan and pazopanib combination demonstrated much higher rates of AEs, which, however, were associated mainly with the classical toxicity of the tyrosine kinase inhibitor. Common adverse events (grade 3 or 4) were fatigue (25%), diarrhea (15%), hand-foot syndrome (10%), mucositis (10%), elevated transaminases (5%), and hypertension (5%) [10]. Dose reduction was required in 70% of patients, but again no patient stopped treatment due to toxicity.

The impact of being overweight or obese on the prognosis of OC is not clear. A recent meta-analysis demonstrated that obesity 5 years before diagnosis or obesity at a young age was associated with poor prognosis [38,39]. However, BMI at diagnosis alone cannot be used as a prognostic factor for the survival of OC patients [39]. It is well known that obese OC patients more often receive lower relative dose intensity of chemotherapy, which significantly impairs their outcomes [40]. However, the impact of BMI > 25 in the studied population was the opposite, with significantly better outcomes observed in overweight/obese patients despite the fact that metronomic chemotherapy dosage was unadjusted to the patient’s weight, and thus the relative dose intensity decreased with increasing BMI. One can speculate that the counterintuitively positive impact of the overweight on the outcome of OC patients receiving metronomic chemotherapy may result from MC’s anti-angiogenic and anti-immunosuppressive activities, which are thought to be dose-independent phenomena [41]. The impact of high BMI on the outcomes of patients treated with MC requires further investigation.

One of the most important limitations of our analysis, besides its retrospective character and relatively low number of patients, is the heterogeneity of the studied population. It is, therefore, difficult to draw any robust conclusions regarding the efficacy of this treatment compared to standard therapeutic options. This is especially true in the case of platinum-sensitive OC patients, in whom the benefit from MC seemed less pronounced than the potential benefit from standard platinum-based therapy. However, those patients who were offered MC and had potentially platinum-sensitive disease were not eligible for platinum-based chemotherapy due to refusal or poor performance status. In general, the MC is assumed as an optimal treatment in the case of asymptomatic or mildly symptomatic patients presenting with slowly progressing disease or as maintenance therapy. The MC may also represent the only applicable option for patients unfit for aggressive, intravenous chemotherapy regimens.

## 5. Conclusions

This is the first report on metronomic chemotherapy based on a topotecan + cyclophosphamide combination in a clinical setting. The CyTo regimen demonstrated a largely acceptable safety profile and promising clinical activity in advanced OC patients. It represents a convenient and relatively inexpensive form of systemic treatment that may be considered an option in asymptomatic or mildly-symptomatic ovarian cancer patients without an imminent threat of visceral crisis. A phase II clinical trial evaluating the combination is warranted.

## Figures and Tables

**Figure 1 cancers-15-01067-f001:**
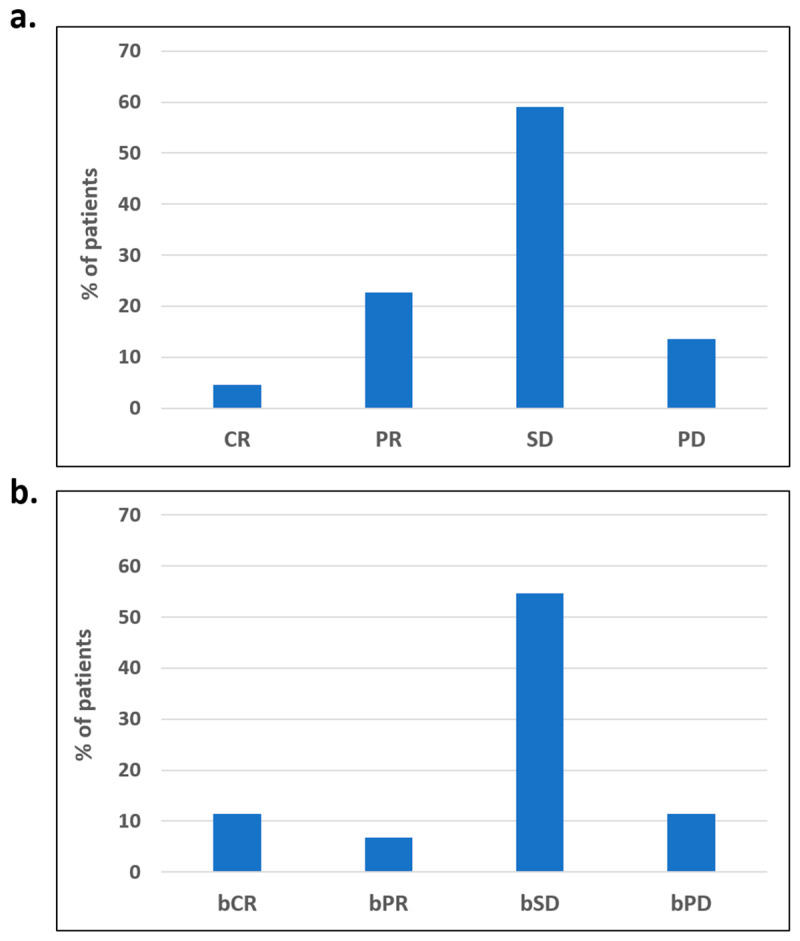
Analysis of responses. (**a**) Responses according to RECIST 1.1 criteria—CR—complete response; PR—partial response; SD—disease stabilization, PD—progressive disease. (**b**) Biochemical responses according to GCIG guidelines—bCR—biochemical complete response; bPR -biochemical partial response; pSD—biochemical stabilization; bPD—biochemical progression.

**Figure 2 cancers-15-01067-f002:**
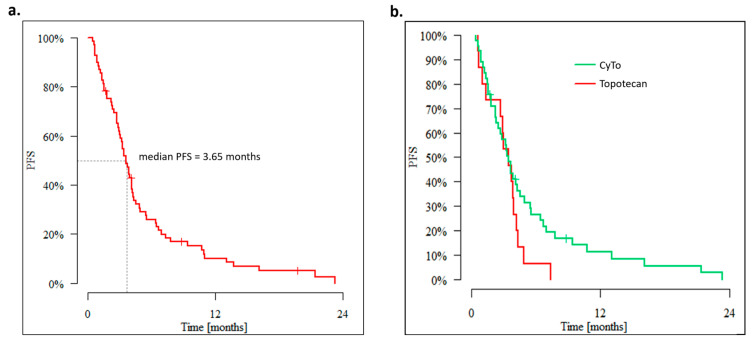
Progression-free survival (**a**) in the whole population; (**b**) in patients treated with single-agent topotecan and topotecan + cyclophosphamide (CyTo) combination.

**Figure 3 cancers-15-01067-f003:**
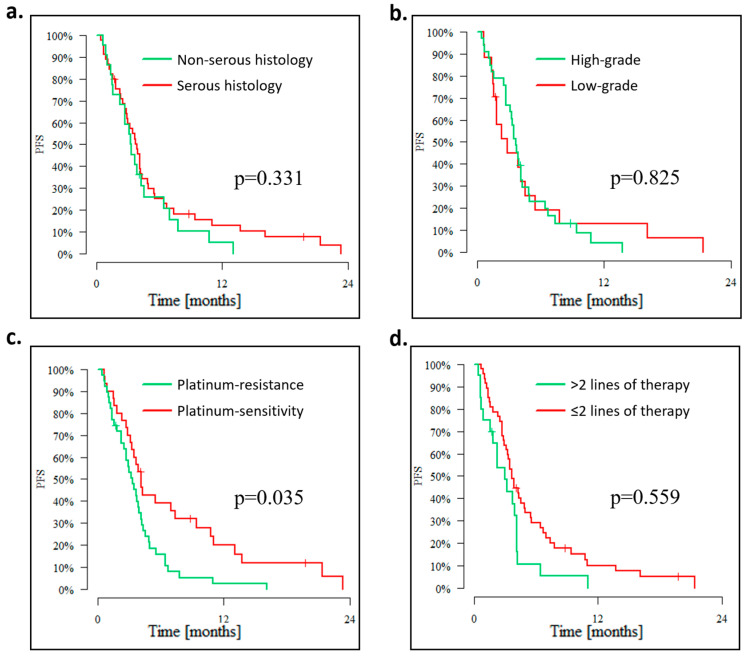
Progression-free survival (**a**) according to histology; (**b**) according to tumor grade; (**c**) according to platinum-sensitivity, (**d**) according to the number of lines of previous systemic treatment.

**Figure 4 cancers-15-01067-f004:**
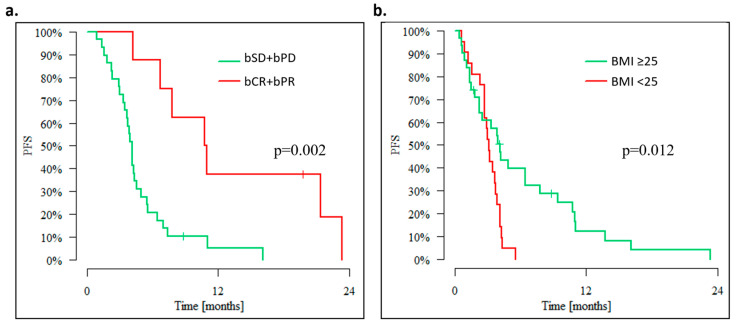
Progression-free survival according to (**a**) biochemical response, (**b**) BMI.

**Table 1 cancers-15-01067-t001:** Clinicopathological characteristics of patients.

		N	%
Population	Total number of patients	72	
Age	Median (years)	60.5	
	<60 years of age	34	47.2%
	≥60 years of age	38	52.8%
Histological type	Serous	48	66.7%
	Other	24	33.3%
Histological grade	Low	17	23.6%
	High	35	48.6%
	Unknown	20	27.8%
Stage at diagnosis (FIGO)	I	5	6.9%
	II	4	5.6%
	III	26	36.1%
	IV	8	11.1%
	Unknown	29	40.3%
Body-Mass Index (BMI)	<25	23	31.9%
	≥25	31	43.1%
	Unknown	18	25.0%
Duration of palliative systemic treatment before initiation of metronomic chemotherapy	≤12 months	24	33.3%
	>12 months	45	62.5%
	Unknown	3	4.2%
Previous lines of chemotherapy	1	17	24.3%
	2	19	27.1%
	3	8	11.4%
	≥4	14	20.0%
Platinum-resistance	Yes	31	43.1%
	No	40	55.6%
	Unknown	1	1.4%
Number of sites with recurrent/metastatic disease	1	4	5.6%
	2	17	23.6%
	3–4	26	36.1%
	≥5	16	22.2%
Metronomic chemotherapy	Topotecan	18	25.0%
	Topotecan → CyTo	9	12.50%
	CyTo	45	62.50%

**Table 2 cancers-15-01067-t002:** Treatment-emergent adverse events in patients receiving metronomic chemotherapy.

	G 1–4		G 3–4	
	N	%	N	%
Any AE	57	89.1%	21	32.8%
neutropenia	35	64.8%	10	18.5%
thrombocytopenia	18	34.6%	2	3.8%
anemia	45	83.3%	14	25.9%
transaminase elevation	21	42.0%	2	4.0%
bilirubine elevation	3	6.0%	1	2.0%
creatinine elevation	16	31.4%	0	0.0%
nausea	9	14.1%	0	0.0%
constipation	4	6.3%	0	0.0%
diarrhea	0	0.0%	0	0.0%

## Data Availability

The data presented in this study are available on request from the corresponding author.

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
