# Peer review of "Metronomic Chemotherapy Based on Topotecan or Topotecan and Cyclophosphamide Combination (CyTo) in Advanced, Pretreated Ovarian Cancer"

_cancers, 2023, doi:10.3390/cancers15041067_

Round 1

Reviewer 1 Report

Wysocki PJ et al presents a retrospective analysis focused on the use of metronomic chemotherapy in advanced ovarian cancer patients. The paper is well written, and each part is clearly explained.

Minor revision:

The only thing I would better clarify is why this cohort of OC patients underwent a metronomic therapy instead of canonical chemotherapy. You should discuss this point in the methods and/or in the discussion.

Author Response

Dear Reviewer,

Thank you for a detailed review of our manuscript entitled "Metronomic chemotherapy based on topotecan or topotecan and cyclophosphamide combination (CyTo) in advanced, pre-treated ovarian cancer."

We appreciate your time and expertise in commenting on our study. We completely agree with your advice to discuss the choice of metronomic instead of the primary, platinum-based chemotherapy in platinum-sensitive OC patients.  

The reason was mentioned in the 'Patients' section of Materials&methods "Metronomic chemotherapy has been initially offered to pretreated, progressing (radiographic and biochemical progression) patients who were deemed unfit or refused further lines of standard intravenous chemotherapy regimens. Subsequently, the treatment was also offered to pretreated, progressing patients who were asymptomatic or mildly symptomatic and were reluctant to initiate intensive, intravenous chemotherapy immediately."

Based on your comment, we decided to thoroughly discuss this issue in the 'Discussion' section.

"

One of the most important limitations of our analysis is the heterogeneity patient population. It isn't easy to draw any robust conclusions regarding the efficacy of this treatment compared to standard therapeutic options. This is especially true in the case of platinum-sensitive OC patients, in whom the benefit from MC seemed less pronounced than the potential benefit from standard platinum-based therapy. However, those patients who were offered MC and had potentially platinum-sensitive disease were not eligible for platinum-based chemotherapy due to refusal or poor performance status. In general, the MC is assumed as an optimal treatment for asymptomatic or mildly symptomatic patients presenting with slowly progressing disease or as maintenance therapy. The MC may also represent the only applicable option for patients unfit for aggressive, intravenous chemotherapy regimens."

Reviewer 2 Report

The article is well written and the issue of metronomic chemotherapy is interesting in highly pretreated ovarian cancer. However, I'm worried about:

- The absence of date from phase I studies regarding the safety and pharmacokinetic of the association of cyclophosphamide and topotecan. How the schedule that the authors used has been discovered?

- The sample size of 72 patients is small and more than 50% of patients have previously received only 1-2 lines of therapy; this is an issue since ovarian cancer is known as a chemoresponsive disease and metronomic chemo is generally  offered in later lines

- 55% of patients were platinum sensitive and have not received platinum. this could justify the response rate that seems high with CyTo

Author Response

Dear Reviewer,

Thank you for a detailed review of our manuscript entitled "Metronomic chemotherapy based on topotecan or topotecan and cyclophosphamide combination (CyTo) in advanced, pre-treated ovarian cancer."

We appreciate your time and expertise in commenting on our study.

The choice of metronomic chemotherapy regimen was based on a few available data on the use of oral metronomic topotecan in cancer patients [Ref 9-11]

↵ Minturn JE, Janss AJ, Fisher PG, Allen JC, Patti R, Phillips PC, Belasco JB: A phase ii study of metronomic oral topotecan for recurrent childhood brain tumors. Pediatr Blood Cancer 56: 39-44, 2011

Tillmanns TD, Stewart CF, MacEachern J, Schaiquevich P, Walker MS, Stepanski EJ: Daily oral topotecan: Utilization of a metronomic dosing schedule to treat recurrent or persistent solid tumors. In: ASCO Annual Meeting: J Clin Oncol, 2008.

Turner, D.C.; Tillmanns, T.D.; Harstead, K.E.; Throm, S.L.; Stewart, C.F. Combination Metronomic Oral Topotecan and Pazopanib: A Pharmacokinetic Study in Cancer Patients with Gynecological Cancer. Anticancer Res 2013, 33, 3823.

We completely agree with your advice to discuss the choice of metronomic instead of the basic, platinum-based chemotherapy in platinum-sensitive OC patients.  

The reason for the utilization of MC instead of standard chemotherapy, even in earlier lines of treatment, was mentioned in the 'Patients' section of Materials&methods "Metronomic chemotherapy has been initially offered to pretreated, progressing (radiographic and biochemical progression) patients who were deemed unfit or refused further lines of standard intravenous chemotherapy regimens. Subsequently, the treatment was also offered to pretreated, progressing patients who were asymptomatic or mildly symptomatic and were reluctant to initiate intensive, intravenous chemotherapy immediately."

Based on your comment, we decided to thoroughly discuss this issue in the 'Discussion' section

"One of the most important limitations of our analysis is the heterogeneity patient population. It isn't easy to draw any robust conclusions regarding the efficacy of this treatment compared to standard therapeutic options. This is especially true in the case of platinum-sensitive OC patients, in whom the benefit from MC seemed less pronounced than the potential benefit from standard platinum-based therapy. However, those patients who were offered MC and had potentially platinum-sensitive disease were not eligible for platinum-based chemotherapy due to refusal or poor performance status. In general, the MC is assumed as an optimal treatment for asymptomatic or mildly symptomatic patients presenting with slowly progressing disease or as maintenance therapy. The MC may also represent the only applicable option for patients unfit for aggressive, intravenous chemotherapy regimens."

We hope our responses and actions will be satisfying and acceptable to you. 

Reviewer 3 Report

Dear author,

very interesting to see data of MC in OC. However, the design of this retrospective analysis is unclear for me. About 50% of the patients have platinum sensitive disease and about 50% just failed 1 or 2 prior lines..

This is a patient population which usually is treated again with platinum. If this is the hypothetical comparator of your MC, the data are really poor.

I think the appropriate patient population for your therapy is platinum resistant disease after failure of at least 2 lines of treatment. 

Could you please limit your analysis to this patient population or describe more clearly the characteristics of the included patients?

Author Response

Dear Reviewer,

Thank you for a detailed review of our manuscript entitled "Metronomic chemotherapy based on topotecan or topotecan and cyclophosphamide combination (CyTo) in advanced, pre-treated ovarian cancer."

We appreciate your time and expertise in commenting on our study. We completely agree with your advice to discuss the choice of metronomic instead of the primary, platinum-based chemotherapy in platinum-sensitive OC patients.  

We completely agree with your remarks on the specific population analyzed in our retrospective analysis. The standard treatment of advanced OC patients in early lines of treatment is intravenous, usually platinum-based chemotherapy. However, the MC was offered only for patients not eligible for standard chemotherapy regimens or for those who refused this type of treatment. This issue was addressed in the 'Patients' section of Materials&methods: "Metronomic chemotherapy has been initially offered to pretreated, progressing (radiographic and biochemical progression) patients who were deemed unfit or refused further lines of standard intravenous chemotherapy regimens. Subsequently, the treatment was also offered to pretreated, progressing patients who were asymptomatic or mildly symptomatic and were reluctant to initiate intensive, intravenous chemotherapy immediately."

Since the number of patients in our retrospective study is relatively low, limiting the population only to platinum-resistant patients who failed >2 lines of treatment would make the analysis almost undoable. Therefore we have decided to stick to the available data and thoroughly discuss the issues raised by you in the 'Discussion' section. 

We hope the extended discussion answers, at least to some extent, your questions and suggestions.

"One of the most important limitations of our analysis, besides its retrospective character and a relatively low number of patients, is the heterogeneity of the studied population. It is, therefore, difficult to draw any robust conclusions regarding the efficacy of this treatment compared to standard therapeutic options. This is especially true in the case of platinum-sensitive OC patients, in whom the benefit from MC seemed less pronounced than the potential benefit from standard platinum-based therapy. However, those patients who were offered MC and had potentially platinum-sensitive disease were not eligible for platinum-based chemotherapy due to refusal or poor performance status. In general, the MC is assumed as an optimal treatment in the case of asymptomatic or mildly symptomatic patients presenting with slowly progressing disease or as maintenance therapy. The MC may also represent the only applicable option for patients unfit for aggressive, intravenous chemotherapy regimens."

Round 2

Reviewer 2 Report

The Authors have improved the strength of their work. The article can now be accepted. I have only one additional request as  I've seen citations that should be improved: lines 236-237 the authors write about the clinical improvement with PARP inhibitors in first line therapy but they cite the SOLO 2 trial (second line) and the PAOLA 1 trial. I suggest to cite some recent summerizing reviews as this one published in JAMA

https://jamanetwork.com/journals/jamanetworkopen/fullarticle/2799126

Author Response

Thank you for the suggestion. The citation of Bartoletti M et al. JAMA  Netw Open 2022 has been included in the discussion section.